# Detection of *NTRK* Fusions and TRK Expression and Performance of pan-TRK Immunohistochemistry in Routine Diagnostics: Results from a Nationwide Community-Based Cohort

**DOI:** 10.3390/diagnostics12030668

**Published:** 2022-03-09

**Authors:** Bart Koopman, Chantal C. H. J. Kuijpers, Harry J. M. Groen, Wim Timens, Ed Schuuring, Stefan M. Willems, Léon C. van Kempen

**Affiliations:** 1Department of Pathology and Medical Biology, University of Groningen, University Medical Center Groningen, P.O. Box 30.001, 9700 RB Groningen, The Netherlands; b.koopman@umcg.nl (B.K.); w.timens@umcg.nl (W.T.); e.schuuring@umcg.nl (E.S.); s.m.willems@umcg.nl (S.M.W.); 2Foundation PALGA, De Bouw 123, 3991 SZ Houten, The Netherlands; chantal.epskamp-kuijpers@palga.nl; 3Department of Pulmonary Diseases, University of Groningen, University Medical Center Groningen, P.O. Box 30.001, 9700 RB Groningen, The Netherlands; h.j.m.groen@umcg.nl

**Keywords:** *NTRK*, TRK, fusion, nationwide, molecular diagnostics, routine, testing

## Abstract

Gene fusions involving *NTRK1*, *NTRK2,* and *NTRK3* are rare drivers of cancer that can be targeted with histology-agnostic inhibitors. This study aimed to determine the nationwide landscape of *NTRK*/TRK testing in the Netherlands and the usage of pan-TRK immunohistochemistry (IHC) as a preselection tool to detect NTRK fusions. All pathology reports in 2017–2020 containing the search term ‘TRK’ were retrieved from the Dutch Pathology Registry (PALGA). Patient characteristics, tumor histology, *NTRK*/TRK testing methods, and reported results were extracted. *NTRK*/TRK testing was reported for 7457 tumors. Absolute testing rates increased from 815 (2017) to 3380 (2020). Tumors were tested with DNA/RNA-based molecular assay(s) (48%), IHC (47%), or in combination (5%). A total of 69 fusions involving *NTRK1* (*n* = 22), *NTRK2* (*n* = 6) and *NTRK3* (*n* = 41) were identified in tumors from adult (*n* = 51) and pediatric (*n* = 18) patients. In patients tested with both IHC and a molecular assay (*n* = 327, of which 29 *NTRK* fusion-positive), pan-TRK IHC had a sensitivity of 77% (95% confidence interval (CI), 56–91) and a specificity of 84% (95% CI, 78–88%). These results showed that pan-TRK IHC has a low sensitivity in current routine practice and warrants the introduction of quality guidelines regarding the implementation and interpretation of pan-TRK IHC.

## 1. Introduction

Gene fusions that involve the neurotrophin tyrosine/tropomyosin receptor kinases 1, 2, and 3 genes (*NTRK1*, *NTRK2,* and *NTRK3*, which encode TRKA, TRKB, and TRKC proteins, respectively) result in constitutively active fusion proteins with an active TRK kinase domain that drives the development of cancer [1]. *NTRK* fusions are pathognomonic in rare types of cancer, such as the *ETV6-NTRK3* fusions in secretory carcinoma of the breast or salivary gland [2,3], congenital mesoblastic nephroma [4], and infantile fibrosarcoma (IFS) [5]. In addition, *NTRK* fusions drive subsets of numerous other tumor types, including Spitzoid neoplasms (10–25%) [6], thyroid cancers (2–3%) [7,8], and rare cases of colorectal cancers (CRC) (<1%), primary brain tumors (<1%), sarcomas (<1%), and non-small cell lung cancers (NSCLC) (<1%) [8].

In recent years, multiple small molecule inhibitors targeting the kinase domain of TRKA, TRKB, and TRKC have been developed and tested in clinical trials. TRK-specific inhibitor larotrectinib and the multi-kinase inhibitor entrectinib (which also targets the tyrosine-protein kinases ALK and ROS) have both yielded remarkable responses in these patients [9,10]. Between 2018 and 2020, these inhibitors received histology-agnostic approvals by both the U.S. Food and Drug Administration (FDA) and the European Medicines Agency (EMA) for adult and pediatric patients with solid cancers harboring fusions in *NTRK1*, *NTRK2,* and *NTRK3* [11,12,13,14].

The availability of effective inhibitors has sparked interest in testing methods that can screen for or confirm the presence of an *NTRK* fusion in tumor tissue. Currently available methods include pan-TRK immunohistochemistry (IHC) [15], and molecular assays such as fluorescence in situ hybridization (FISH) [16], fusion gene-specific reverse-transcriptase polymerase chain reaction (RT-PCR) [17], targeted DNA- or RNA-based sequencing [18], and molecular RNA counting [19]. International guidelines from both the American Society of Clinical Oncology and the European Society for Medical Oncology agree that molecular assays are necessary to show the presence of the actual *NTRK* fusion, and pan-TRK IHC may be used as a screening tool [20,21,22,23]. The latter recommendation was based on three studies that reported a sensitivity of 95–100% [15,24,25]. However, more recent studies report a lower sensitivity (75–88%) for pan-TRK IHC, especially for *NTRK3* (55–79%) [26,27].

The objective of this study was to determine the testing landscape and incidence of *NTRK* fusions reported in routine diagnostics in the Netherlands and to determine the concordance between IHC and molecular assays.

## 2. Materials and Methods

### 2.1. Patient Selection

The nationwide network and registry of histo- and cytopathology in the Netherlands (PALGA, Houten, The Netherlands) maintain a database with excerpts of all pathology reports from pathology departments in the Netherlands [28]. For this study, all pathology reports of which the microscopy and/or conclusion text contained the search term ‘TRK’ between 2017 and 2020 were retrieved. These reports were screened for assays testing TRK expression and/or the presence of *NTRK* fusions. Patients were excluded if the test failed or if the result was not reported (Appendix A). The data request was approved by the scientific and privacy committee of PALGA (application number LZV2019-119) and made accessible in accordance with General Data Protection Regulation (EU) 2016/679.

### 2.2. Data Extraction and Handling

Variables were extracted from the pathology reports by a dedicated researcher (BK) and included patient age and sex, tumor histology, mutations that were detected in other (driver) genes, whether TRK expression or *NTRK* fusions were, the type of test(s) used to test for TRK/*NTRK*, and the reported TRK expression and/or *NTRK* fusion status. Data processing was performed in accordance with the General Data Protection Regulation (EU) 2016/679.

### 2.3. Interpretation of Reported TRK/NTRK Testing Results

The reported TRK/*NTRK* testing results were interpreted in accordance with (inter) national consensus guidelines [20,21,22,23]. These state that a positive or inconclusive IHC result should be confirmed with a molecular assay such as FISH or an RNA analysis (Appendix A). If all test results were negative, it was interpreted as “no *NTRK* fusion.” A positive molecular result was interpreted as “*NTRK* fusion positive.” In case of a positive or inconclusive IHC staining, the result was interpreted as “unconfirmed” when no subsequent confirmatory molecular assay was performed and “*NTRK* fusion positive” if a subsequent molecular assay did demonstrate presence of an *NTRK* fusion. If a molecular test yielded an inconclusive result, or if there were discrepancies between two or more molecular tests, the result was interpreted as “inconclusive”.

### 2.4. Statistics

Statistical analysis was performed with SPSS version 23 (SPSS Inc., Chicago, IL, USA). Descriptive statistics were used, proportional differences were assessed using Fisher’s exact tests. The performance of IHC as a screening tool in the routine setting for the detection of *NTRK* fusions was determined by calculating sensitivity and specificity for positive IHC results with the molecular result as a reference. This calculation was performed for cases with only positive IHC and repeated for cases with either positive or inconclusive results. A nominal *p*-value of 0.05 or less was considered statistically significant.

## 3. Results

### 3.1. Patients Included in the Analysis 

A total of 14,025 pathology reports contained the search term ‘TRK’ in PALGA between 2017 and 2020, representing 9595 unique malignant and benign tumors. TRK expression and/or *NTRK* fusion testing results were reported in 7457 tumors from 7434 patients (for 23 patients, two unique tumors were tested for *NTRK*). The number of tested tumors gradually increased from 2017 (*n* = 815) to 2020 (*n* = 3380) (Figure 1A).

### 3.2. TRK Expression or NTRK Fusion Testing: Types of Tumors and Reasons for Testing

Lung cancer was the most frequent type of tumor tested for *NTRK* (*n* = 3625; 48.6%). Less frequent types of tumors included soft tissue/bone tumors (*n* = 835; 11.2%), melanocytic tumors (*n* = 662; 8.9%), cancers of unknown primary (*n* = 486; 6.5%), CRC (*n* = 328; 4.4%), thyroid tumors (*n* = 245; 3.3%) and salivary gland tumors (*n* = 215; 2.9%), among others (Figure 1B). Tumor biopsies were tested for TRK expression or *NTRK* fusions for choice of therapy (*n* = 5147; 69.0%), differential diagnosis (*n* = 2056; 27.6%) or resistance analysis after therapy progression (*n* = 250; 3.4%) (Figure 1C). Indications that proportionally increased in 2019–2020 as opposed to 2017–2018 were the differential diagnosis of melanocytic tumors (increase from 0.5% to 12.1%; *p* < 0.001) and resistance analysis at progression on targeted therapy (increase from 1.1% to 4.2%; *p* < 0.001). The latter category included 248 lung cancer patients progressing on ALK, BRAF or EGFR inhibitors and two patients with a gastro-intestinal stromal tumor progressing on imatinib.

### 3.3. Assays Used to Test for TRK Overexpression and NTRK Fusions

Tumors were tested with a molecular analysis only (*n* = 3587; 48.1%), IHC only (*n* = 3496; 46.9%), IHC and a molecular analysis (*n* = 352; 4.7%), or with an unspecified test (*n* = 22; 0.3%) (Figure 1D). In total, 3848 tumors were tested with IHC, but for 2536 tumors (66%), the clone was not specified. The pan-TRK rabbit monoclonal antibodies specified in reports were EPR17341 (Abcam, Cambridge, MA, USA; *n* = 655) and A7H6R (Cell Signaling Technology, Danvers, MA, USA; *n* = 701; Figure 1E). The EPR17341 antibody was only used as part of a laboratory-developed test: the recently approved CE-IVD kit with this antibody (Ventana Medical Systems, Oro Valley, AZ, USA) was not reported.

FISH analysis was the most used DNA-based molecular test (277/280) (Figure 1F). The proportion of tumors tested only with a DNA assay decreased in 2019–2020 as opposed to 2017–2018 (4.8% to 1.4%), whereas the proportion tested with only multiplex RNA analysis increased (35% to 49%). Nearly all (>98%) of RNA-based molecular assay (*n* = 3715) were multiplex RNA analyses, and the vast majority of tumor biopsies were tested with Archer FusionPlex kits (Archer DX, Boulder, CO, USA; *n* = 3273) (Figure 1G).

Of note, the FDA-approved companion diagnostic FoundationOne CDx was not used by any laboratory. This is in part because this assay was approved only at the end of the inclusion period of the current study (October 2020) [29]. In addition, Dutch and European guidelines currently do not specify which RNA assay is most suitable for the detection of *NTRK* fusions [21,22].

### 3.4. NTRK Fusions or TRK Expression Detected in Routine Diagnostics

Between 2017 and 2020, a total of 7457 tumors (7434 patients) were tested for TRK expression and/or *NTRK* fusions, and an *NTRK1-3* fusion was detected with a molecular assay in 69 (0.93%) tumors (Table 1 and Appendix A). This proportion was 1.8% when counting only the patients tested with a molecular assay (*n* = 3939). These percentages do not reflect the true prevalence of *NTRK* fusions in the population as there is a selection bias in routine diagnostics (patients will only be tested when clinically relevant). For this reason, frequencies could not be calculated for individual tumor types. In addition, there were nine patients with inconclusive molecular results or with discrepancies between two or more molecular tests.

The detected *NTRK* fusions involved *NTRK1* (*n* = 22), *NTRK2* (*n* = 6) and *NTRK3* (*n* = 41). Frequently identified *NTRK* fusion genes included *TMP3-NTRK1* (*n* = 7), *LMNA-NTRK1* (*n* = 4), *ETV6*-*NTRK3* (*n* = 28), and *MYO5A*-*NTRK3* (*n* = 4). In addition to these four fusions, 18 other unique fusion genes were reported (Figure 2A). *NTRK* fusion-positive patients had varying types of malignant and benign tumors (Figure 2B).

In addition to the 69 molecular-positive patients, there were 130 patients (1.7%) with a positive (*n* = 67) or inconclusive IHC result (*n* = 63) for whom molecular assay was not performed to confirm (or exclude) the presence of an *NTRK* fusion. Reasons for not confirming the IHC result varied but were not specified in the vast majority of cases (*n* = 113; 87%).

### 3.5. IHC as a Preselection Tool to Detect Fusions

A total of 352 tumors were tested with both IHC and a molecular assay. After excluding patients for whom either test failed (*n* = 15), with inconclusive molecular results (*n* = 4) or insufficient information (*n* = 6), 327 were eligible for a sensitivity/specificity analysis, of which 29 patients had a molecular-confirmed *NTRK* fusion. There were 114 patients with either a positive (*n* = 61) or inconclusive (*n* = 53) IHC result. An *NTRK* fusion was confirmed in 20% (23 of 114); the positivity rate was 33% (20 of 61) for IHC-positive patients and 6% (3 of 53) for IHC-inconclusive patients. On the other hand, there were six patients who tested negative with IHC, but who nevertheless harbored an *NTRK* fusion using a molecular test (representing 21% of all *NTRK* fusion-positive patients tested with both IHC and a molecular assay). This included multiple well-described *NTRK3* fusions, including *ETV6-NTRK3*, *MYO5A-NTRK3*, and *TPM3-NTRK1*. These six patients with IHC-negative *NTRK* fusion-positive tumors were tested in five different laboratories, and the antibody clone was only reported in one (EPR17431, laboratory-developed). Pan-TRK IHC was found to have a sensitivity of 77% (95% CI, 56–91) and a specificity of 83% (95% CI, 78–88%) when excluding patients with an inconclusive IHC result. When patients with an inconclusive IHC result were also included, sensitivity raised slightly to 79% (95% CI, 60–92%), but specificity declined to 70% (95% CI, 64–74%) (Table 2).

## 4. Discussion

In this study, the incidence of *NTRK* fusions in tumors was investigated in the Dutch population between 2017 and 2020 reported in the nationwide PALGA database of pathology reports. Over time, tumor biopsies were increasingly tested for TRK expression or *NTRK* fusions, with most tumors being tested with IHC and/or multiplex RNA analysis. In this four-year period, 69 *NTRK* fusion-positive tumors were reported in 7434 patients tested. In addition, the data were used to assess the performance of pan-TRK IHC as a pre-screening tool to detect *NTRK* fusions. The analysis of real-world data indicated a low sensitivity for pan-TRK IHC to preselect patients for molecular *NTRK* testing in routine practice.

### 4.1. Testing Rates for TRK Expression and NTRK Fusion in The Netherlands

Testing for *NTRK* fusions was not standard-of-care for any type of cancer until 2020, when *NTRK* became part of the required molecular markers that any advanced NSCLC had to be tested for according to the Dutch guideline for NSCLC [30]. However, *NTRK* fusions have been of interest since 2017, as diagnostic markers of disease (for example, for secretory carcinoma) or as predictive markers of response to TRK inhibitors, which were then available within clinical trials and in named patient programs. The FDA and EMA have since approved both entrectinib and larotrectinib for the treatment of advanced cancers with an *NTRK* fusion [11,12,13,14]. This is reflected by the increasing national testing rate (from 815 tumors in 2017 to 3380 tumors in 2020), resulting in an increase from 10 *NTRK* fusions diagnosed in 2017 (1.2% of tumors tested) to 31 *NTRK* fusions diagnosed in 2020 (0.9% of tumors tested). As larotrectinib and entrectinib have been approved for use in the Netherlands recently, the testing rate will likely further increase in the coming years.

### 4.2. Types of Tumors Harboring NTRK Fusions

*NTRK* fusions are highly enriched in secretory carcinomas (of the salivary gland or breast), congenital mesoblastic nephroma, and IFS (>90%) [2,3,4,5]. In the current dataset, these cancers were all reported, with *NTRK* fusions detected in similar proportions, though the absolute numbers were relatively low. For example, there were only two *NTRK* fusion-positive secretory carcinomas of the breast, whereas the expected number in the Netherlands between 2017–2020 is approximately 10–11 (estimated incidence 0.015% of approximately 17,000 annual breast cancer diagnoses) [31,32,33]. The low incidence in the current dataset is likely because these cancers are not yet routinely tested for the presence of *NTRK* fusions but rather are diagnosed clinicopathologically and/or immunophenotypically, or that patients often present with early-stage disease and are therefore not eligible for treatment with TRK inhibitors [32].

Spitzoid melanocytic tumors also commonly harbor *NTRK* fusions (2–25%) [6,7,8]. Indeed, these tumors were among the most reported *NTRK* fusion-positive tumors in the Dutch population (*n* = 10). In Spitzoid tumors, *NTRK1* and *NTRK3* fusions have frequently been reported, and these fusions are a diagnostic marker for this category of melanocytic tumors [34,35]. *NTRK1* and *NTRK3* fusions were indeed more common in the Dutch population (*n* = 4 and *n* = 5, respectively), mostly in benign tumors, and all of these patients were tested for the differential diagnosis. Of note, one patient with a benign Spitz nevus harbored a previously reported *SQSTM1*-*NTRK2* fusion, which confirms findings from a recent case report that Spitz/Reed nevi can harbor fusions in any of the three *NTRK* genes [36].

In thyroid tumors, *NTRK* fusions are also relatively common (2–25%) [7,8], though some studies report that this is mostly true for papillary thyroid carcinoma (PTC) [37]. In the Dutch population (13 *NTRK* fusion-positive thyroid cancers), PTC was indeed the most frequent subtype of *NTRK* fusion-positive thyroid cancer (6/13), but *NTRK* fusions were also detected in follicular variants of PTC, a Hürthle cell carcinoma, and a poorly differentiated thyroid carcinoma. Thus, the histological subtype of thyroid cancer does not exclude the presence of an *NTRK* fusion. 

In addition, *NTRK* fusions are rare (<1%) drivers of other types of cancer, such as NSCLC, CRC, primary brain tumors, and soft tissue/bone tumors [8]. In NSCLC and CRC, the prevalence is estimated at 0.23% [38,39]. In our dataset, a similar prevalence of 0.25% was found in NSCLC (9/3625), representing fusions in all three *NTRK* genes. In CRC, 89% of *NTRK* fusion-positive patients have mismatch repair deficiency (MMRd) and lack mutations in *BRAF* (*BRAF*wt); when testing only MMRd/*BRAF*wt CRC patients, the prevalence increases from 0.23% to 5.3% [40]. In our dataset, *NTRK* fusion-positive CRC was not identified despite a sample size of 328 patients. 

Furthermore, *NTRK* fusions were detected in varying types of non-IFS soft tissue/bone tumors (*n* = 14), of which the majority (*n* = 8) were classified as *NTRK*-rearranged spindle cell neoplasm/sarcoma in accordance with the 2020 World Health Organization Classification of Tumors of Soft Tissue and Bone [41]. However, *NTRK* fusion-positive soft tissue/bone tumors diagnosed prior to 2020 (which include solitary fibrous tumors, an angiofibroma, and chondrosarcoma) may have been classified similarly had they been diagnosed according to the 2020 WHO classification. Considering *NTRK* fusions have been recommended by the WHO guideline as diagnostic markers for sarcomas, the annual incidence of *NTRK* fusion-positive sarcomas is expected to further increase in the coming years.

### 4.3. Fusion Partners of NTRK1–3 

A range of different *NTRK1*, *NTRK2* and *NTRK3* fusions were reported in the Dutch population between 2017 and 2020. A total of 22 unique RNA-confirmed fusion genes were identified. *ETV6*-*NTRK3* fusions, which are among the most well-described *NTRK* fusions in human tumors [3], were the most common in this study (41%; 28/69). This prevalence was higher than reported in other studies (26–29%) [42], and may be due to an overrepresentation as this has been a known diagnostic marker and thus was the only fusion for which a targeted RT-PCR was used. Aside from *ETV6-NTRK3*, 10 other fusion genes identified in our study had been previously reported: *CD74*-*NTRK1* [43], *LMNA*-*NTRK1* [44], *TPM3*-*NTRK1* [44], *TPR*-*NTRK1* [45], *NACC2*-*NTRK2* [46], *STRN*-*NTRK2* [47], *SQSTM1*-*NTRK2* [48], *EML4*-*NTRK3* [49], *MYH9*-*NTRK3* [35], *MYO5A*-*NTRK3* [35]. Three other fusions were novel, although the fusion partner has been reported in other driver genes: *GRIPAP1*-*NTRK1* (fusion partner previously reported for *TFE3*) [50], *TRIM24*-*NTRK2* (*BRAF*) [51], and *BEND5*-*NTRK2* (*ALK*) [52]. The remaining eight fusion partners were novel: *EFNA1*-*NTRK1*, *GP2*-*NTRK1*, *PRG4*-*NTRK1*, *FMN2*-*NTRK2*, *CDH2*-*NTRK3*, *CNTN4*-*NTRK3*, *SYNJ1*-*NTRK3*, and *UBE2F*-*NTRK3*. Although all these fusions harbored the TRK kinase domain (Figure 2A), their pathogenicity is not determined. Tumors harboring *EFNA-NTRK1* and *CNTN4-NTRK3* fusions also had a positive pan-TRK IHC, and tumors harboring the *GP2-NTRK1* fusion had both positive IHC and FISH results, which reinforces the notion that these fusions result in increased expression of the TRK kinase domain. In contrast, the tumor harboring a *CDH2-NTRK3* fusion, which had a large portion of intact TRKC protein (fused at exon 7 of TRKC), had no TRK expression with IHC. The other novel fusions were not tested with IHC.

### 4.4. Co-Occurrence of Other Driver Mutations

Previous studies have demonstrated that *NTRK* fusions are typically mutually exclusive with other driver mutations [16,26]. However, studies have reported co-occurrence of *NTRK* fusions with well-known drivers such as oncogenic *BRAF*, *EGFR,* or *KRAS* mutations [53,54]. In our dataset, 53 of the 69 *NTRK* fusion-positive patients were tested for concurrent driver mutations and/or fusions (Appendix A), of which all but three patients did not harbor another driver mutation. Two of these three patients, all of whom had lung adenocarcinoma, were treatment naïve. One *EGFR* p.(L858R)-mutant patient harbored the novel *PRG4*-*NTRK1* fusion (TRK expression was not tested), and one a *KRAS* p.(A146T)-mutant patient harbored a *CD74*-*NTRK1* fusion, with TRK expression. The third patient, who harbored an *EGFR* p.(E746_A750del) mutation, acquired the novel *EFNA1*-*NTRK1* fusion as a resistance mechanism against EGFR inhibitor treatment; in this patient, TRK expression was confirmed with IHC. This finding is in line with previous reports that *NTRK* fusions can induce resistance to EGFR inhibitors [55]. It is unknown how co-occurrence of *NTRK* fusions with other driver mutations would impact TRK inhibitor sensitivity.

### 4.5. Performance of Immunohistochemistry as a Screening Tool for the Detection of NTRK Fusions

Guidelines and consensus recommendations indicate that the presence of *NTRK* fusions can be most reliably detected using molecular assays [20,21,22,23]. In addition, they suggest that IHC is a reliable screening tool to exclude the presence of *NTRK* fusions. In general, IHC is of interest as a screening tool because it is relatively inexpensive, efficient, and easily implemented in pathology laboratories [21]. For some markers, such as ALK, IHC has proven to be a valuable alternative to predict molecular marker-based treatment response [56], whereas, in other markers, such as MET [57], IHC expression correlates poorly with the presence of a molecular deficit. For a screening tool to qualify as useful, its ability to exclude the presence of a condition, its sensitivity, needs to be near to 100% [58]. However, a high specificity is also desirable to ensure the cost-effectiveness of the test (the lower the specificity, the more patients will require both the screening test and the confirmatory test). Guidelines recommending pan-TRK IHC as a screening tool were mostly based on three studies from 2017 and 2018 (Table 3) [15,24,25]. However, subsequently, two more recent studies demonstrated lower sensitivity to screen for *NTRK* fusions, especially for *NTRK3* [26,27], which was reflected in the current nationwide study based on reports from routine testing. Multiple TRK IHC-negative cases with well-described *NTRK3* fusions, including *ETV6-NTRK3* and *MYO5A-NTRK3* were reported.

It is uncertain whether these fusions really do not lead to IHC positivity or whether the IHC procedure or the choice of antibodies requires optimization and/or standardization. For the predictive biomarker PD-L1, it was demonstrated previously that preanalytical variability in immunostaining procedures could lead to inter-rater variability in scoring among pathologists [59]. Similarly, there are currently no quality guidelines regarding preanalytical requirements for immunostaining with pan-TRK IHC antibodies. Introducing such quality guidelines may improve the sensitivity of pan-TRK IHC to pre-screen tumors for the presence of *NTRK* fusions.

A multitude of antibodies are available, but most laboratories use a pan-TRK antibody, which targets the C-terminus of TRKA, TRKB, and TRKC [60]. Two antibodies are utilized especially: EPR17341 and A7H6R. It has been demonstrated that EPR17341 and A7H6R result in highly different staining patterns [61]. In our study, the antibody used to test for TRK expression was not reported in the majority of reports. The EPR17341 antibody is the only antibody that is available in a CE-IVD-approved kit but demonstrated insufficient sensitivity in the studies by Solomon et al. (88%) and Galatica et al. (75%). A false-negative IHC result was also identified in the current dataset (in melanoma harboring a *TPM3-NTRK1* fusion), which indicates that the current immunohistochemical procedure may not be adequate to screen for *NTRK* fusions. The A7H6R antibody is not available in a CE-IVD-approved kit and has not been thoroughly investigated in the literature. Future studies should elucidate whether this antibody is more sensitive, and/or focus on the development of novel antibodies that may have improved sensitivity compared to the currently available antibodies.

There are no standardized scoring criteria to determine the positivity of TRK IHC, including for the CE-IVD approved assay [62]. In the current study, this resulted in a nearly equal number of positive and inconclusive patients. Pathologists reported an inconclusive result when they were uncertain whether the percentage of cells with positive staining or the intensity of staining could be classified as positive. Indeed, in 53 tumors with an inconclusive IHC that were also tested with a molecular assay, only three (6%) were found to harbor an *NTRK* fusion. In addition, depending on the fusion present, TRK IHC can stain differently (cytoplasmic, perinuclear, nuclear, or membranous) [15]. This underscores the necessity to develop internationally standardized criteria for scoring TRK IHC positivity, which is currently lacking.

### 4.6. Future Role of pan-TRK Immunohistochemistry in Diagnostic Algorithms

Targeted RNA analysis is considered the gold standard for detecting *NTRK* fusions, and all international guidelines recommend confirmation of positive pan-TRK IHC with a targeted RNA analysis [20,21,22,23]. The findings in this as well as other recent studies that address sensitivity and specificity of pan-TRK IHC [26,27], indicate that up-front testing with a targeted RNA analysis should be preferred in some scenarios. In case of a high prevalence of *NTRK* fusions in cancer types with a low incidence, such as the *ETV6-NTRK3* fusions in secretory carcinomas, using a suboptimal pre-screening tool that requires confirmation is not practical. This is also true for tumors with high endogenous TRK expression, such as gliomas and some types of soft tissue tumors. Pre-screening with pan-TRK IHC will yield a high rate of false-positivity. Up-front RNA testing may also be feasible for patients with a frequently occurring type of cancer and a low prevalence of *NTRK* fusions. For example, patients with NSCLC are routinely tested for the presence of fusions in *ALK*, *RET*, *ROS1,* and *MET* exon 14 skipping in a single RNA-based assay that can often easily be expanded with the *NTRK1–3* fusions. In spite of a 100% sensitivity of TRK IHC in the five lung cancer cases in the current study, it is more efficient from a tissue management and cost perspective to test NSCLC patients with an up-front, panel-based, targeted RNA assay rather than adding an IHC screening for TRK expression. For some types of cancer, such as CRC and non-Spitz melanoma, it is not cost-effective to test all patients using a costly RNA-based technique as less than 1% actually harbor an *NTRK* fusion, and they are not routinely tested using an RNA-based analysis for other reasons. For these types of cancer, a pre-screening test such as pan-TRK IHC may still have its value when procedures (analytical and interpretation) are optimized.

Altogether, the results of the current study support findings from other recent studies that pan-TRK IHC with currently used antibodies is not yet adequate to screen tumors for the presence of *NTRK* fusions in routine diagnostics. Quality guidelines regarding the implementation of validated pan-TRK IHC antibodies and standardized staining and scoring criteria are required.

## 5. Conclusions

This nationwide, longitudinal analysis shows that routine TRK testing is gradually increasing, with *NTRK* fusions detected in a variety of 69 benign and malignant tumors in 2017–2020. A performance analysis demonstrated low sensitivity and specificity of immunohistochemistry as a preselection tool for the detection of *NTRK* fusions in current routine practice. Recommendations by international guidelines that pan-TRK IHC is a reliable screening tool should, therefore, be reconsidered in light of the current study and recently published real-world evidence. These results warrant the introduction of quality guidelines regarding the implementation of validated pan-TRK IHC antibodies and standardized staining and scoring criteria for interpreting results in a routine diagnostic setting. More sensitive pan-TRK antibodies may be needed. 

## Figures and Tables

**Figure 1 diagnostics-12-00668-f001:**
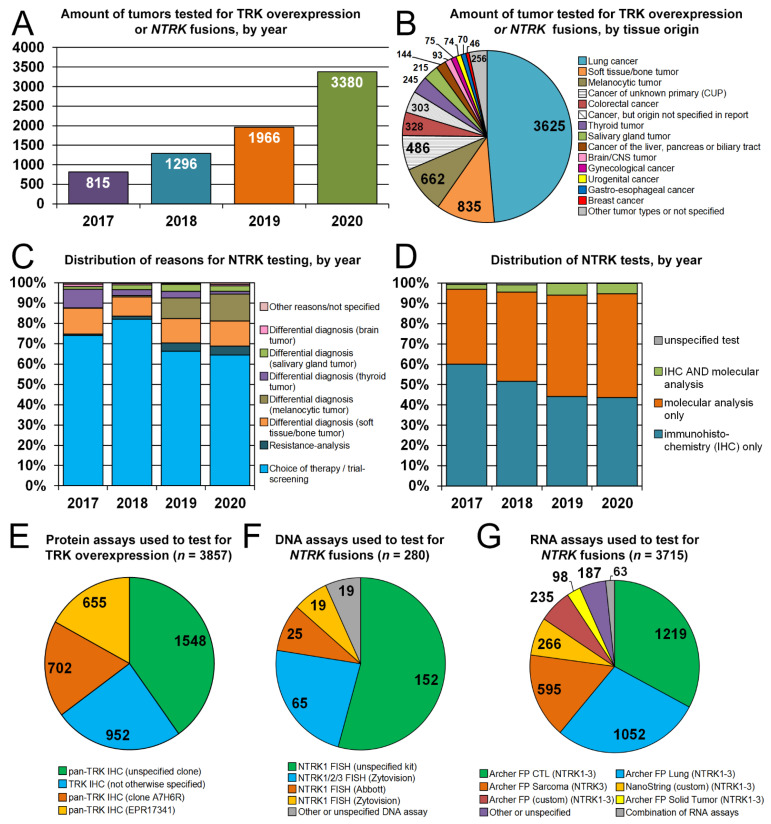
Indications and number of TRK overexpression and/or *NTRK* fusions testing in the Netherlands, 2017–2020. (**A**,**B**) Number of tumors tested for TRK overexpression or *NTRK* fusions, by year (**A**) and tissue origin (**B**). (**C**) Distribution of reasons for *NTRK*/TRK testing, by year. (**D**) Distribution of *NTRK* tests, by year. (**E**–**G**) Protein (**E**), DNA (**F**) and RNA (**G**) assays used to test for TRK overexpression or *NTRK* fusions. Abbreviations: CNS, central nervous system; CTL, comprehensive thyroid and lung; FISH, fluorescence in situ hybridization; FP, FusionPlex.

**Figure 2 diagnostics-12-00668-f002:**
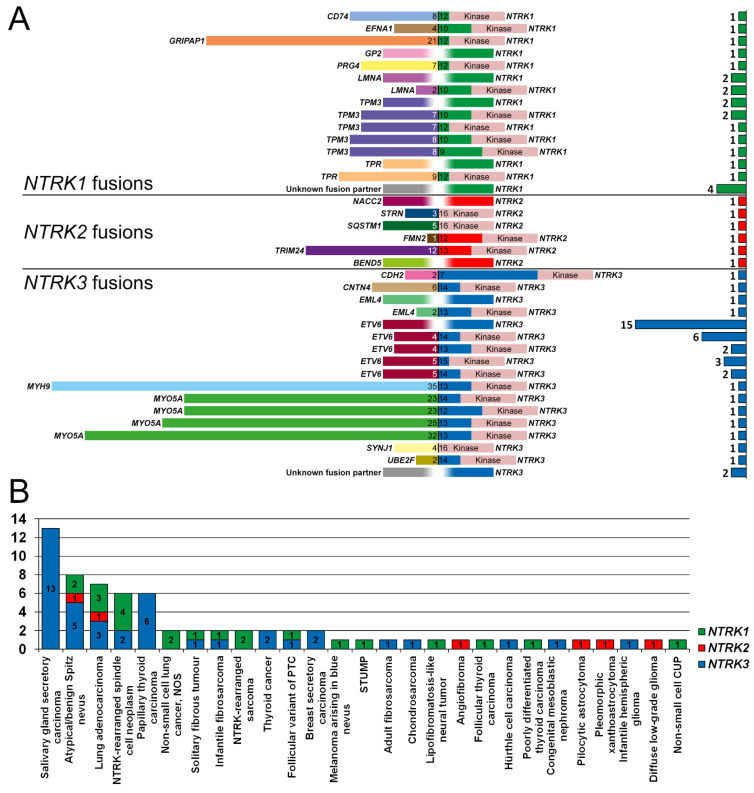
*NTRK1-3* fusions identified in the Dutch population, 2017–2020. (**A**) Fusion partners and exonic breakpoints were identified in 69 patients with *NTRK1* (green), *NTRK2* (red), and *NTRK3* (blue) fusions. The kinase domain is annotated and shown in pink. A blurred white breakpoint indicates that the exact breakpoint was not known or specified. At the right, a bar graph depicts the number of times the specific fusion gene was detected. (**B**) Frequency of *NTRK1* (green), *NTRK2* (red), and *NTRK3* (blue) fusions and types of malignant or benign tumors. Abbreviations: CUP, cancer of unknown primary; NOS, not otherwise specified; PTC, papillary thyroid carcinoma, STUMP, Spitzoid Tumor of Uncertain Malignant Potential.

**Table 1 diagnostics-12-00668-t001:** Characteristics of patients with a molecular-confirmed *NTRK1-3* fusion.

Characteristic	All (*n* = 69)	Adults (≥18 years) (*n* = 51)	Children (<18 Years) (*n* = 18)
Sex			
Male, *n* (%)	32 (46.4)	23 (45.1)	9 (50.0)
Female, *n* (%)	37 (53.6)	28 (54.9)	9 (50.0)
Age			
Mean (range)	37.7 (0–84)	48.5 (19–84)	7.0 (0–16)
Year tested			
2017, *n* (%)	10 (14.5)	9 (17.6)	1 (5.6)
2018, *n* (%)	8 (11.6)	7 (13.7)	1 (5.6)
2019, *n* (%)	20 (29.0)	15 (29.4)	5 (27.8)
2020, *n* (%)	31 (44.9)	20 (39.2)	11 (61.1)
Tumor type			
Soft tissue/bone tumor, *n* (%)	16 (23.2)	9 (17.6)	7 (38.9)
Salivary gland secretory carcinoma, *n* (%)	13 (18.8)	13 (25.5)	–
Thyroid tumor, *n* (%)	13 (18.8)	10 (19.6)	3 (16.7)
Melanocytic tumor, *n* (%)	10 (14.5)	6 (11.8)	4 (22.2)
Lung cancer, *n* (%)	9 (13.0)	9 (17.6)	–
Brain/CNS tumor, *n* (%)	4 (5.8)	1 (2.0)	3 (16.7)
Breast secretory carcinoma, *n* (%)	2 (2.9)	2 (3.9)	–
Congenital mesoblastic nephroma, *n* (%)	1 (1.4)	–	1 (5.6)
Cancer of unknown primary, *n* (%)	1 (1.4)	1 (2.0)	–
Reason for testing *NTRK*			
Differential diagnosis, *n* (%)	55 (79.7)	39 (76.5)	16 (88.9)
Choice of therapy/trial-screening, *n* (%)	13 (13.8)	11 (21.6)	2 (11.1)
Resistance-analysis after progression, *n* (%)	1 (1.4)	1 (2.0)	–
Type of test used			
Multiplex RNA analysis, *n* (%)	25 (36.2)	19 (37.3)	6 (33.3)
IHC and multiplex RNA analysis, *n* (%)	18 (26.1)	12 (23.5)	6 (33.3)
Fusion gene-specific RT-PCR, *n* (%)	10 (14.5)	10 (19.6)	–
IHC, FISH and multiplex RNA analysis, *n* (%)	8 (11.6)	6 (11.8)	2 (11.1)
IHC and unspecified molecular assay, *n* (%)	2 (2.9)	1 (2.0)	1 (5.6)
FISH and multiplex RNA analysis, *n* (%)	2 (2.9)	–	2 (11.1)
Unspecified, *n* (%)	2 (2.9)	1 (2.0)	1 (5.6)
FISH, *n* (%)	1 (1.4)	1 (2.0)	–
IHC and ISH, *n* (%)	1 (1.4)	1 (2.0)	–

Abbreviations: FISH, fluorescence in situ hybridization; IHC, immunohistochemistry; RT-PCR, real-time polymerase chain reaction.

**Table 2 diagnostics-12-00668-t002:** Sensitivity and specificity of pan-TRK IHC as a screening tool for *NTRK* fusions with molecular assay as a reference.

Pan-TRK IHC	Molecular Assay			
Only positive IHC	Negative	Positive	Total	
Negative	207	6	213	NPV (95% CI): 97.2% (94.5–98.6%)
Positive	41	20	61	PPV (95% CI): 32.8% (25.6–40.9%)
Total	248	26	274	
	Specificity (95% CI): 83.5% (78.3–87.9%)	Sensitivity (95% CI): 77.0% (56.4–91.0%)		
Including inconclusive IHC	Negative	Positive	Total	
Negative	207	6	213	NPV (95% CI): 97.2% (94.4–98.6%)
Positive/inconclusive	91	23	114	PPV (95% CI): 20.2% (16.4–24.6%)
Total	298	29	327	
	Specificity (95% CI): 69.5% (63.9–74.6%)	Sensitivity (95% CI): 79.3% (60.3–92.0%)		

Abbreviations: CI, confidence interval; IHC, immunohistochemistry; NPV, negative predictive value; PPV, positive predictive value.

**Table 3 diagnostics-12-00668-t003:** Sensitivity of TRK immunohistochemistry as a screening tool to detect *NTRK* fusions, according to published studies.

Study	Sensitivity (*n*) ^a^							
	*NTRK1*		*NTRK2*		*NTRK3*		Overall	
Hechtman 2017 [15]	100	(10/10)	100	(2/2)	88.9	(8/9)	95.2	(21/22)
Murphy 2017 [25]	Affected *NTRK* genes not specified						100	(8/8)
Rudzinski 2018 [24]	100	(12/12)	100	(1/1)	94.1	(16/17)	96.7	(29/30)
Solomon 2019 [26]	96.2	(26/27)	100	(5/5)	79.4	(27/34)	87.9	(58/66)
Gatalica 2019 [27]	87.5	(7/8)	88.9	(8/9)	54.5	(6/11)	75.0	(21/28)
Current study	86	(13/15)	100	(2/2)	67	(8/12)	79.3	(23/29)

^a^ For each gene, the first column displays sensitivity (in %), and the second column displays the absolute division (a/b) used to calculate sensitivity, using the number of patients with *NTRK* fusions (b) that tested positive with TRK immunohistochemistry (a).

## Data Availability

All data generated or analyzed during this study are included in this published article [and its Appendix A].

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
