# Peer review of "Detection of NTRK Fusions and TRK Expression and Performance of pan-TRK Immunohistochemistry in Routine Diagnostics: Results from a Nationwide Community-Based Cohort"

_diagnostics, 2022, doi:10.3390/diagnostics12030668_

Round 1

Reviewer 1 Report

In this manuscript, Koopman et al. perform a retrospective analysis of all pathology reports in the Dutch Pathology Registry from 2017-2020 to analyze and compare different methodologies for NTRK fusion detection.   The study is well-designed, well-presented and well-written.  The findings generally agree with previously published studies.  The low sensitivity and specificity of the pan-TRK antibody is an important conclusion and is appropriately emphasized.  The findings of this study and their discussion are extremely relevant for pathologists in a modern practice.  

I want to note that I enjoyed the extremely thorough discussion and especially appreciated the discussion of the future of diagnostic algorithms. 

No comments or questions.

Author Response

Reviewer 1:

No comments or questions.

Author’s response: We thank the reviewer kindly for his/her comment and for taking the time to consider our manuscript.

Reviewer 2 Report

This original study analyzed the nationwide registry database with excerpts of all pathology reports from pathology departments in the Netherlands to obtain the real-world data for NTRK fusion testing, testing results, and sensitivity and specificity of NTRK IHC (immunohistochemistry). The reviewer thinks that the data of this study is unique, the analysis was well-performed, and the results of this study is clinically important. The reviewer has only a few minor comments.

  1. Table ST1 shows that there is no IHC negative case in lung cancer patients with NTRK fusions. Because lung cancer patients have quite low incidence of NTRK fusion, but lung cancer itself is a very common disease, the reviewer thinks it would be helpful if NTRK IHC screening is effective at least in lung cancers. Could the authors add some comments, considering the past publications (ref. 15, 24-27), regarding this point?
  2. The results of Foundation One CDx seems not to be included in this database (Figure 1G). Is there any reason for this? 

Author Response

Reviewer 2:

  1. Table ST1 shows that there is no IHC negative case in lung cancer patients with NTRK fusions. Because lung cancer patients have quite low incidence of NTRK fusion, but lung cancer itself is a very common disease, the reviewer thinks it would be helpful if NTRK IHC screening is effective at least in lung cancers. Could the authors add some comments, considering the past publications (ref. 15, 24-27), regarding this point?

Author’s response: Indeed, there were five NTRK fusion-positive lung cancer patients tested with both IHC and a molecular assay, and all of these demonstrated TRK IHC positivity. This suggests that the low sensitivity of TRK IHC may not apply to lung cancer and that NTRK IHC may therefore be an effective screening tool. However, patients with NSCLC are routinely tested for the presence of fusions in ALK, RET, ROS1 and MET exon 14 skipping in a single RNA-based assay that can often easily be expanded with the NTRK1-3 fusions. As such, in spite of the higher sensitivity of IHC in lung cancer, we believe it is more efficient from a tissue management and cost perspective to test NSCLC patients with an up-front, panel-based, targeted RNA assay rather than adding an IHC screening for TRK expression. We have adjusted the Discussion section to address this point (lines 377-380).

  1. The results of Foundation One CDx seems not to be included in this database (Figure 1G). Is there any reason for this?

Author’s response: The reviewer is right to point out that FoundationOne CDx is currently the only FDA-approved companion diagnostic for the detection of NTRK fusions. However, the approval of this assay was only provided in October 2020, which is at the end of our four-year study period. In addition, Dutch laboratories do not adhere to the FDA, but rather to its Dutch and European counterparts, which currently do not recommend a specific assay for the detection of NTRK fusions. We have clarified this in the section 3.3 of the Results (lines 137-141).